# Computer-Aided Prediction, Synthesis, and Characterization of Magnetic Molecularly Imprinted Polymers for the Extraction and Determination of Tolfenpyrad in Lettuce

**DOI:** 10.3390/foods12051045

**Published:** 2023-03-01

**Authors:** Du Chi, Wei Wang, Shiyin Mu, Shilin Chen, Kankan Zhang

**Affiliations:** 1National Key Laboratory of Green Pesticide, Guizhou University, Guiyang 550025, China; 2Key Laboratory of Green Pesticide and Agricultural Bioengineering, Ministry of Education, Guizhou University, Guiyang 550025, China; 3Center for R&D of Fine Chemicals, Guizhou University, Guiyang 550025, China

**Keywords:** tolfenpoyrad, molecular imprinting technology, analytical method, magnetic molecularly imprinted polymer

## Abstract

Tolfenpyrad, a pyrazolamide insecticide, can be effectively used against pests resistant to carbamate and organophosphate insecticides. In this study, a molecular imprinted polymer using tolfenpyrad as a template molecule was synthesized. The type of functional monomer and the ratio of functional monomer to template were predicted by density function theory. Magnetic molecularly imprinted polymers (MMIPs) were synthesized using 2-vinylpyridine as a functional monomer in the presence of ethylene magnetite nanoparticles at a monomer/tolfenpyrad ratio of 7:1. The successful synthesis of MMIPs is confirmed by the results of the characterization analysis by scanning electron microscopy, nitrogen adsorption–desorption isotherms, Fourier transform infrared spectroscopy, X-ray diffractometer, thermogravimetric analyzer, and vibrational sample magnetometers. A pseudo-second-order kinetic model fit the adsorption of tolfenpyrad, and the kinetic data are in good agreement with the Freundlich isothermal model. The adsorption capacity of the polymer to the target analyte was 7.20 mg/g, indicating an excellent selective extraction capability. In addition, the adsorption capacity of the MMIPs is not significantly lost after several reuses. The MMIPs showed great analytical performance in tolfenpyrad-spiked lettuce samples, with acceptable accuracy (intra- and inter-day recoveries of 90.5–98.8%) and precision (intra- and inter-day relative standard deviations of 1.4–5.2%).

## 1. Introduction

Tolfenpyrad is a pyrazolamide insecticide and acaricide developed by Mitsubishi Chemical Corporation of Japan. Its mechanism of action is to inhibit complex I in the mitochondrial respiratory electron transport chain [1]. Because of its universal insecticidal activity, tolfenpyrad is widely used in vegetables, fruits, and other crops [2]. However, long-term use of pesticides inevitably causes people to worry about the residues of green leafy vegetables and the environment, which may pose a potential threat to human health [3]. Therefore, it is necessary to develop a sensitive and simple detection method. Several methods for the determination of tolfenpyrad have been reported, including liquid chromatography with mass spectrometry (LC-MS) [4], gas chromatography with tandem mass spectrometry (GC-MS/MS) [5], and liquid chromatography with tandem mass spectrometry (LC-MS/MS) [2]. While these methods can effectively identify tolfenpyrad, these samples contain a large number of complex matrices and the analysis of tolfenpyrad often requires complex sample preparation steps. As instrumental analysis technology cannot directly adapt to complex samples, the pretreatment process of sample preparation is considered to be the most critical step in the entire analysis process, and the method of sample preparation determines the quality of results [6].

Molecular imprinting technology (MIT), as an approach for synthesizing molecularly imprinted polymers (MIPs), is a relatively new pretreatment technology for pesticide residue analysis, overcoming the limitations of traditional pretreatment methods [7]. MIPs can be interpreted as synthetic analogues of natural biological antibody–antigen systems. It works by using a “lock and key” mechanism to selectively bind the molecules in which they are templated in the production process. MIPs have specific recognition sites, which can specifically combine with target compounds called templates in shape, size, and function, and then extract target compounds from different matrices [8,9]. MIPs are tailored to target analytes; even in complex environments and matrices, they can efficiently adsorb and elute specific target compounds to achieve the goal of target compound separation and enrichment. Among them, the pretreatment method of magnetic solid phase extraction (MSPE) using magnetic nanoparticles (MNPs) as the extraction medium has received extensive attention [10,11,12]. MSPE has efficient dispersion and rapid circulation capacity, and has a large dispersion surface volume ratio, large extraction capacity, and easy separation of adsorbent using an external magnetic field [13]. Therefore, MSPE is often used to capture target analytes in the pretreatment of pesticide residue analysis.

In the synthesis of MIPs, the function of functional monomers is to form pre-polymerization complexes with templates by providing functional groups. Therefore, the most important task is to find appropriate functional monomers, so that they can have the maximum interaction with the template in the ground state, thus obtaining high selectivity and rebinding ability [14]. Conventional MIT often optimizes the synthesis conditions by repeated experiments, which leads to problems of long experimental periods and substantial material consumption. The computer molecular simulation technology can play a predictable guiding role in the preparation process, and can realize the tailoring of precise recognition sites and the design of recognition driving force [15]. The stability of the recognition system is optimized through the calculation of physical and chemical characteristic parameters such as binding energy, so as to reasonably select template molecules, functional monomers, cross-linking agents, and pore forming agents; optimize polymerization conditions; improve the specificity and affinity of polymer recognition; and shorten the experimental cycle [16]. The computer-aided prediction of MIPs based on density function theory (DFT) has been applied to select functional monomers of specific template molecules, which has been confirmed and widely used [17,18]. In contrast to the traditional synthesis and application of MIPs, the development of magnetic molecular imprinting materials with the aid of DFT allows for the selective identification of pesticides with rapid separation efficiency from crop samples.

In this paper, we develop and validate an MIT method with LC-MS/MS for the determination of tolfenpyrad. Gaussian 09W software was used to screen for appropriate functional monomers and the ratio of functional monomers to template molecules. Fe_3_O_4_-MNPs were synthesized by the coprecipitation method. Magnetic molecularly imprinted polymers (MMIPs) were prepared using tolfenpyrad as a template molecule, divinylpyridine (2-VP) as a functional monomer, and ethylene glycol dimethacrylate (EGDMA) as a crosslinking agent. Then, the polymer structure was characterized by X-ray diffractometer (XRD), thermogravimetric analyzer (TGA), Fourier transform infrared spectroscopy (FT-IR), specific surface area test (BET), scanning electron microscopy (SEM), and vibrating sample magnetometer (VSM). The adsorption properties of the polymers were evaluated using both dynamic and thermodynamic adsorption experiments. Several other structural analogs have been used to verify the selectivity of MMIPs against tolfenpyrad. This approach not only shortens the sample pretreatment duration, but also accurately and efficiently detects the residues of tolfenpyrad in the actual sample.

## 2. Materials and Methods

### 2.1. Reagents and Chemicals

Standard of tolfenpyrad (95%, purity) was purchased from Heowns Biochem Technologies, LLC, Tianjin (Tianjin, China). Analytical grade iron (II) chloride tetrahydrate (FeCl_2_·4H_2_O) and iron (III) chloride tetrahydrate (FeCl_3_·6H_2_O) were obtained from Tianjing Chemical Reagent Company (Tianjin, China). Analytical grade ammonium hydroxide (25%), vinyltrimethoxysilane (VTMS), EGDMA, 2-VP, ethanol, methanol (MeOH), acetonitrile (ACN), 4,4′-azobis(4-cyanovaleric acid), N,N-dimethylformamide (DMF), and acetic acid (AA) were bought from Chengdu Jinshan Chemical Reagent Co., Ltd. (Chengdu, China).

### 2.2. Instrumentation

A SU8100 SEM (Hitachi Ltd., Tokyo, Japan) was used to confirm the morphology of the polymers. A D8 Advance XRD (Bruker Corporation, Billerica, MA, USA), recording in the 2θ range of 20–80° at a speed of 5°/min under CuKα radiation (*λ* = 0.15418 nm), was applied on the determination of crystal structure. A Nicolet iS5 FT-IR spectrometer (Thermo Fisher Scientific Inc., Waltham, MA, USA; spectra scanning range, 500–4000 cm^−1^; mode: transmission; resolution, 4 cm^−1^) was used to analyze the functional groups. A Setline STA8000 TGA (KEP Technologies, Lyon, France; temperature range, 25–500 °C; heating rate, 10 °C/min) was used to determine the mass loss of curves. Moreover, a Lake Shore 7404 VSM (Lake Shore Cryotronics, Inc., Westerville, OH, USA) and an ASAP2460 BET (Micromeritics Instrument Corporation, Norcross, GA, USA) were applied to investigate other characteristics of tolfenpyrad polymers. A digital stainless steel hotplate magnetic stirrer (J&K Scientific Ltd., Beijing, China), a DZF-6050 vacuum drying oven (Shanghai scientific instrument Co., Ltd., Shanghai, China), a BSD-YX2200 vertical type intelligent precision shaker (Shanghai Boxun Medical Biological Instrument Corporation, Shanghai, China), a MTV-100 multi-tube vortexer (Hangzhou Allsheng Instrusment Co., Ltd., Hangzhou, China), and a BILON10-300C ultrasonic cleaning machine (Shanghai Bilon Instrument Manufacturing Co., Ltd., Shanghai, China) were used for the synthesis and adsorption experiments of tolfenpyrad polymers. Chromatographic analysis was performed on a Shimadzu LC-20AD XR LC system (Shimadzu Corporation, Kyoto, Japan) coupled with a Sciex API 4000 Q-TRAP quadrupole MS system (Applied Biosystems, Foster City, CA, USA). Chromatographic separation was conducted using an Agilent Eclipse XDB-C_18_ column (4.6 × 150 mm, 5 µm, Agilent Technologies, Foster City, CA, USA). The mobile phases were acetonitrile and 0.1% FA aqueous solution (90/10, *v*/*v*). Then, 5 µL of sample was injected at a flow rate of 0.8 mL/min. The elution time was about 6 min. The gas source in all cases was nitrogen (99.9999%, purity). Positive electron spraying ionization (ESI^+^) and multiple reaction monitoring (MRM) modes were chosen for analysis. The respective quantitation and confirmation transitions are m/z 384.4→197.3.0 and m/z 384.4→145.2. The declustering potential is 110.8 V and the collision energies are 19.74 V for quantitation and 14.47 V for confirmation transitions. Other MS parameters include the curtain gas pressure, 172 kPa; ion source gas 1 and 2 pressures, 414 kPa; ion source temperature, 600 °C; and ion spray voltage, 5.5 kV.

### 2.3. Computational Simulation

To select the best functional monomers and the appropriate ratio of functional monomers to template molecules, we chose Gaussian 09W software for the simulation. B3LYP, PBEPBE, and ωB97XD methods coupled with 6–31G (d, p) and 6–31G+ (d, p) basis sets were employed to optimize the structure, frequency, and stability energies [19]. The interaction energy (Δ*E*) of tolfenpyrad with a monomer was calculated according to the following equation:(1)$$\Delta E=E_{P}-E_{T}-E_{M}$$
where *E_P_* is the energy of the template–monomer complex after structure optimization and *E_T_* and *E_M_* are the energies of the template molecule and monomer, respectively.

### 2.4. Synthesis of Vinylized MNPs and MMIPs

Triiron tetraoxide nanoparticles were prepared by the co-precipitation method (Figure 1) reported by Kong et al. [20]. FeCl_3_·6H_2_O (2.701 g) and FeCl_2_·4H_2_O (0.994 g) were dispersed in 50 mL of ultrapure water and regularly stirred at 300 rpm for 20 min under constant N_2_ purging. The samples were transferred to a silicon oil bath whose temperature was raised to 70 °C under continuous purging and stirring. Then, 30 mL of 28% ammonium hydroxide (*w*/*w*) was dropwise added over 30 min and refluxed for 1 h. The resulting black particles were collected with an external magnetic field and ultrasonically washed with high-purity water to a neutral pH, and then dried in a vacuum oven at 60 °C overnight. Vinylization of MNPs was conducted following the method described by Shao et al. [21]. Here, 0.5 g of the MNPs was added into ethanol (125 mL) and the mixture was sonicated for 15 min. After adding 4 mL of ammonia and 4 mL of VTMS, the mixed solution was mechanically agitated overnight at room temperature under an N_2_ air flow. Finally, the vinylized magnetite nanoparticles (VMNPs) were washed with ethanol and dried at 60 °C for 2 h.

Tolfenpyrad (0.2 mmol) and 2-VP (1.4 mmol) were dispersed in DMF (10 mL) under sonication, and the mixed solution was prepolymerized at 4 °C for 12 h. Subsequently, the vinylated MNPs were ultrasonically dispersed in 10 mL of DMF, followed by the addition of the prepolymerization solution, EGDMA (4 mmol), and 4,4′-azobis(4-cyanovaleric acid) (50 mg) in sequence. The mixture was purged with N_2_ for 20 min and stirred for 120 min at 80 °C. The obtained MMIPs were washed with methanol to remove the residual reagents, and then eluted using a Soxhlet apparatus with methanol/acetic acid (9:1, *v*/*v*) for 36 h and dried under vacuum. Magnetic non-imprinted polymers (MNIPs) were developed following the same procedure without the presence of template molecules.

### 2.5. Adsorption Isotherm and Kinetic Experiments

The adsorption properties of the MMIPs on tolfenpyrad were investigated by adsorption isotherm experiment. Here, 20 mg of MMIPs was dispersed in 10 mL of ACN containing tolfenpyrad (10, 20, 40, 60, 80, 100, 200, and 300 mg/L). The solution was sonicated for 3 min and then left for 5 h at 25 °C. The concentration of unabsorbed tolfenpyrad was determined using the supernatant by LC-MS/MS. The adsorption capacity *Q* (mg/g) of the MMIPs was calculated by Equation (2) [13]:(2)$$Q=\left(C_{0}-C_{e}\right)V/m$$
where *C*_0_ and *C_e_* are the initial and equilibrium tolfenpyrad concentrations (mg/L), respectively; *V* (mL) is the total volume of the adsorption mixture; and *m* is the mass of the polymer used for each binding mixture.

At present, the main mathematical models for evaluating the affinity of MIPs are Langmuir and Freundlich [10]. According to Shimizu theory, the Langmuir equation belongs to the discrete mathematical model and is suitable for evaluating monolayer adsorption systems with homogeneous solid surfaces, while the Freundlich equation belongs to the continuous mathematical model and is more suitable for heterogeneous solid–liquid adsorption systems. The static adsorption curve of the Langmuir and Freundlich models (Equations (3) and (4), respectively) was described as follows:(3)CeQe=1KLQmax+CeQmax
(4)logQe=logKF+1nlogCe
where *C_e_* is the equilibrium concentration (µg/mL), *Q_e_* is the adsorption capacity (mg/g) at equilibrium, *K_L_* is the Langmuir isotherm constant related to the affinity of the binding sites (L/mg), *Q_max_* is the maximum adsorption capacity (mg/g), *K_F_* is the equilibrium constants relating to adsorption capacity (L/mg), and *n* is the intensity of the Freundlich model.

To investigate the adsorption dynamic kinetics of MMIPs and MNIPs, 20 mg of MMIPs (or MNIPs) was weighed into a 50 mL conical flask and 10 mL of tolfenpyrad (100 mg/L) standard solution was added, then it was shaken at 25 °C for 5–180 min. Afterwards, the supernatant was removed and filtered through a nylon filter. The concentration of tolfenpyrad in the supernatant was determined by LC-MS/MS. The adsorption dynamic curves of pseudo-first-order (Equation (5)) and pseudo-second-order (Equation (6)) models were described as follows [20]:(5)$$\ln\left(Q_{e}-Q_{t}\right)=\ln Q_{1c}-k_{1}t$$
(6)tQt=1k2Q2c2+tQ2c
where *Q_e_* is the adsorption amount of the polymer in equilibrium (mg/g), *Q_t_* is the adsorption amount (mg/g) at time *t* (min), *Q_c_* is the theoretical adsorption amount (mg/g), *K*_1_ is the first-order adsorption rate coefficients (min^−1^), and *K*_2_ is the second-order adsorption rate coefficients (min^−1^).

### 2.6. Selectivity and Reusability of MMIPs

The selectivity of the polymers used for tolfenpyrad was tested using a rebinding experiment. Here, 20 mg of MMIPs (or MNIPs) was suspended in 10 mL of ACN containing 100 mg/L of tolfenpyrad and other two pesticides (penthiopyrad and mandipropamid) and shaken at 25 °C for 70 min. Then, an external magnet was used to separate the supernatant and polymers. The treated solution was filtered with a microporous membrane and detected by LC-MS/MS for the non-extracted amount of each analyte. The selectivity of MMIPs and MNIPs for tolfenpyrad was compared to its structural analogs, two of which are known as penthiopyrad and mandipropamid. The imprinting factor (*IF*) and selectivity coefficient (*SC*) were selected to measure the competitive selectivity [20,22].
(7)IF=QMMIPQMNIP
(8)SC=IFtIFa
where *Q_MMIP_* is the MMIPs’ adsorption capacities (mg/g), *Q_MNIP_* is the MNIPs’ adsorption capacities (mg/g), *IF_t_* is the imprinting factors of the template, and *IF_a_* is the imprinting factors of the structural analogues.

To study the stability and reusability of MMIPs, 20 mg of MMIPs was added into 10 mL of tolfenpyrad ACN standard solution (100 mg/L). The mixture was oscillated at 25 °C and 180 rpm for 70 min, and the MMIPs were separated by an external magnetic field. The supernatant was obtained via microporous membrane filtration, and the concentration of tolfenpyrad was quantified by LC-MS/MS. Soxhlet extraction was used with a mixed solution of methanol and AA (9/1, *v*/*v*) for 36 h for the desorption of tolfenpyrad. The eluted material was dried in a vacuum oven for 10 h at 60 °C. The same adsorption experiment was repeated seven times to observe the change in the adsorption effect of MMIPs.

### 2.7. Application of MMIPs in the Spiked Lettuce Samples

To check the feasibility and application potential of the proposed method, real vegetables were analyzed. The lettuce samples were purchased from a local supermarket in Guiyang, China. The lettuce sample was cut into pieces and homogenized. Then, 10 g of lettuce sample was weighed in a 50 mL centrifuge tube, followed by the addition of 20 mL of different tolfenpyrad ACN standard solution (concentration: 0.05, 0.1, and 1 mg/L), 4 g of anhydrous MgSO_4_, and 2 g of NaCl. The mixture was vortexed at a relative centrifugal force (RCF) of 1677× *g* for 3 min and centrifuged at 4025× *g* for 5 min. The upper ACN layer was further treated with MMIPs and the procedure was same as the adsorption experiment. Finally, the supernatant was filtered through a microporous membrane and detected by LC-MS/MS.

## 3. Results

### 3.1. Selection of Functional Monomers

The theoretical and experimental values of the infrared adsorption wavenumbers of tolfenpyrad are provided in Appendix A. The values obtained on the PBEPBE with 6–31G+ (d, p) basis set are very close to the measured values. Thus, the PBEPBE with 6–31G+ (d, p) basis set was adopted for the further selection of the functional monomer. In our study, 4-vinylaniline (4-VA), 2-VP, methacrylic acid (MAA), and 4-vinylbenzaldehyde (4-VB) were chosen as potential monomers to prepare MIPs. As shown in the spatial structure and molecular electrostatic potential (MEP) values in Figure 2, the dominant imprinted binding sites of the tolfenpyrad template are the O_29_, N_34_, and H_26_ atoms. For monomers, the main active sites of 2-VP are N_6_ on the pyridine ring and H_15_ on the vinyl group, while the main active sites of MAA are O_3_ and H_4_ atoms on the carboxyl group and those of 4-VB are O_13_ and H_7_ atoms. For 4-VA, the main active sites are the N_16_ and H_17_ atoms. The comparison of energy between the template and functional monomers is mentioned in Table 1. As shown, the Δ*E* value of tolfenpyrad-2-VP is the lowest among all Δ*E* values, which indicates that the tolfenpyrad-2-VP complex is more stable than the tolfenpyrad-4-VA, tolfenpyrad-MAA, and tolfenpyrad-4-VB complexes. As shown in Appendix A, 2-VP as the functional monomer will have more interaction forces than the optimized structure of the other three monomers. Moreover, we further screened the ratio of template molecules to functional monomers. From Table 2, the overall energy decreases gradually with the increase in functional monomers. The above results show that 2-VP is a suitable monomer for preparing tolfenpyrad-2-VP, with an optimal ratio of 1:7.

### 3.2. Characterization of the Imprinted Polymers

The SEM images are shown in Appendix A. The surface of MMIPs is rougher than that of MNIPs. XRD analysis was performed on Fe_3_O_4_, vinylized Fe_3_O_4_, and MMIP samples to verify the core–shell structure. In the 2θ region of 20–80°, six characteristic diffraction peaks corresponding to the (220), (311), (400), (422), (440), and (511) planes were found in all three samples (Figure 3A). FT-IR measurements were used to evaluate the accuracy of the synthesis of MMIPs (Figure 3B). For Fe_3_O_4_, the absorption band at approximately 582 cm^−1^ corresponded to the stretching vibration of Fe–O. The modified magnetic core by VTMS induced a peak at 1100 cm^−1^ attributed to the stretching bonds of Si–O–Si and a peak at 1620 cm^−1^ attributable to C=C stretching vibrations. Finally, the absorption peak of the C=O stretching band was found at approximately 1720 cm^−1^ in MMIPs. In Figure 3C, similar shapes and trends of the three magnetization curves and no remanence or coercivity were observed, which may be because the curves passed through the origin. The saturation magnetizations (*M_s_*) of Fe_3_O_4_, vinylized Fe_3_O_4_, and MMIPs were 65.5, 54.6, and 31.40 emu/g, respectively. Figure 3D shows that Fe_3_O_4_ loses only 4% of its weight over the whole temperature range, which is caused by the surface water loss. The weight loss rate of vinylized Fe_3_O_4_ and MMIPs is fast at 240–400 °C, because the organic matter on the surface of the magnetic carrier is decomposed by heat, and the weight loss is about 6% and 20%. The textural properties of the polymers were investigated by nitrogen adsorption–desorption measurements. The data (Figure 4) indicated that the specific surface area of MMIPs (162.90 m^2^/g) was higher than that of MNIPs (98.39 m^2^/g). The pore diameter and pore volume of MMIPs (8.72 nm and 0.3751 nm) were larger than those of the corresponding MNIPs (8.53 nm and 0.2207 nm).

### 3.3. Adsorption Assays

#### 3.3.1. Static Adsorption

Adsorption isotherm was evaluated for both MMIPs and MNIPs over a range of concentration from 0 to 300 mg/L, and the data are presented in Figure 5A. The adsorption increases with concentration and reaches equilibrium at a concentration of 100 mg/L of tolfenpyrad. The results in Figure 5B,C indicate that the static adsorption isotherms of MMIPs and MNIPs are better fit by the Freundlich isothermal model. Moreover, the data in Appendix A indicated that the correlation coefficients (*R*^2^) of two polymers’ isotherms according to the Freundlich equation (Equation (3), 0.992–0.997) were higher than those calculated by the Langmuir equation (Equation (4), 0.924–0.944). The values of *Q_max_* and *K_F_* are 8.65 mg/g and 0.227 L/mg, respectively.

#### 3.3.2. Dynamic Adsorption

The kinetic adsorption curves in Figure 5D demonstrate that the adsorption of MMIPs increases fast before 60 min and reaches the adsorption equilibrium at 70 min, with a saturated adsorption capacity of 7.20 mg/g. The data in Figure 5E,F show that the pseudo-second-order kinetics model has better linearity and higher accuracy than the pseudo-first-order kinetics model. Moreover, in Appendix A, *Q*_2*c*_ was closer to *Q_e_* than *Q*_1*c*_.

### 3.4. Selectivity and Stability

The selectivity results of MMIPs and MNIPs are shown in Figure 6A and Table 3. The *IF* value for tolfenpyrad (8.25) was higher than those for its structural analogs (2.17–3.10), indicating that MMIPs have the strongest adsorption capacity for tolfenpyrad. To measure the stability of the MMIPs, seven regeneration cycles were performed with the standard solution of tolfenpyrad. In Figure 6B, after seven regeneration cycles, the adsorption capacity of the MMIPs is slightly reduced, with a loss of 7%.

### 3.5. Applicability of the Developed Method

The developed method, a combination method of MIT pretreatment and LC-MS/MS detection, was validated and applied on the determination of tolfenpyrad in lettuce samples. The linearity of the established method was evaluated in the linear range of 0.005–1 mg/L. The results showed that the calibration curve of peak area–concentration was satisfactory (*R*^2^ = 0.998). Based on the respective signal-to-noise ratio (*S*/*N*) of 3:1 and 10:1, the limit of detection (LOD) and limit of quantification (LOQ) of the developed approach were calculated as 1.7 and 5 µg/kg, respectively. The accuracy and precision data for method validation are shown in Table 4. The intra-day recoveries of tolfenpyrad in spiked lettuce samples at three concentrations of 5, 10, and 100 µg/kg were 90.5–98.8%, with intra-day relative standard deviations (RSDs) of 1.4–5.2%, and the inter-day recoveries ranged from 90.6% to 98.3%, with inter-day RSDs of 1.7–4.1%.

## 4. Discussion

The infrared adsorption wavenumbers of tolfenpyrad were examined using quantum mechanics of DFT/B3LYP, PBEPBE, and ωB97XD methods with 6–31G (d, p) and 6–31G+ (d, p) basis sets to study the compound [23]. Theoretically, MEP is an important parameter to evaluate the imprinted binding sites in a molecule. The higher the MEP difference between different atoms in a molecule, the more imprinted binding sites are contained in the molecule [19]. The predicted sites can be used as a basis for modeling functional monomers and template molecules. Theoretically, the low Δ*E* value indicates the stable complex among template monomers, and this functional monomer is suitable for imprinted specific templates. When the tolfenpyrad-2-VP ratio is changed from 1:6 to 1:7, the decrease in binding energy is most significant (−325.317380 Hartree), which demonstrates that the stability of the complex has been greatly enhanced at the ratio of 1:7 [24]. SEM was used to characterize the surface morphology of both MMIPs and MNIPs prepared by the surface imprinting polymerization method. The SEM figures present that the addition of template molecules changes the microstructure of MIPs and creates a cavity that can increase the adsorption capacity and improve the mass transfer rate of released and rebound analytes [25]. The XRD results are well matched with the magnetite data from the JCPDS International Center for Diffraction Data (JCPDS Card: 19–629) [26,27]. The modification of the structure of Fe_3_O_4_ with vinyltrimethoxysilane and 2-VP and the formation of MMIPs did not lead to core–shell structural variations. The FT-IR results confirmed the introduction of the vinyl group to the Fe_3_O_4_ nanoparticles [28,29] and demonstrated that the imprinted polymer layer had been satisfactorily prepared on magnetic particles [30]. The magnetic susceptibility of the synthesized material is a crucial aspect in the magnetic separation process, thus magnetic hysteresis loops were obtained with a vibrating sample magnetometer. The MMIP samples exhibited superparamagnetism. The decrease in the magnetization values of MMIPs can be attributed to the screening effect of the silica coating layer and the formation of the imprinted shell. In any case, this value was high enough for the complete magnetic separation in a short time (15 s) using an external magnet. The thermal stability of imprinted polymers on magnetic microspheres was determined by TGA. The TGA results indicate that the thermal properties of vinylized Fe_3_O_4_ are relatively stable at 30–500 °C and the quality of vinylized Fe_3_O_4_ is not significantly lower than that of Fe_3_O_4_. MIPs were successfully polymerized on the surface of vinylized Fe_3_O_4_ with a high grafting efficiency. This is because the Fe_3_O_4_ surface was successfully grafted with vinyl, which can be easily chemically modified and facilitates the coating of imprinted polymer shells on its surface. The above results demonstrate the successful generation of selective cavities in MMIPs.

Compared with MNIPs, MMIPs exhibit high adsorption capacity because of their selective imprinting structure. The values of *Q_max_* calculated from the Freundlich equation are closer to the experimental values. The static adsorption results showed that the Freundlich isothermal model was more suitable for this study, indicating that MMIPs consisted of heterogeneous binding sites [31]. The adsorption behavior of MNIPs is essentially similar to that of MMIPs; however, the adsorption capacity is significantly lower. The adsorption rate of MMIPs is 2.76 times higher than that of MNIPs at 70 min. The reason for this may be the presence of specific recognition sites on MMIPs, which make it easier for tolfenpyrad to enter and exit. The adsorption dynamic kinetic results show that the mechanism of adsorption dynamics of MMIPs is consistent with the pseudo-second-order kinetic model, because the chemical interaction forces between the molecules are the main cause of the adsorption rate. Compared with other amine compounds (such as penthiopyrad and mandipropamid), MMIPs could produce a specific adsorption effect on tolfenpyrad due to the formation of a certain structure and size of pores on the polymers. Higher *SC* values contribute to the high affinity of MMIPs. In summary, the high *IF* values of MMIPs in pesticide molecules with similar structures indicate a specific adsorption capacity for tolfenpyrad. The data demonstrate the good feasibility of this MIT method for the determination of tolfenpyrad.

Re-adsorption of the target molecule into the MIP can be either at molecular recognition sites in the conformation of the MIP (specific adsorption) or nonspecific adsorption on the MIP polymer surface (nonselective) [32,33,34,35]. The interaction assumed in the computer calculation (Appendix A) applies to both the molecular recognition site and the nonselective adsorption on the surface. The good selectivity (Figure 6) assumed that the target molecule was incorporated into the molecular recognition site in the synthesized MMIP. The recognition sites of the MMIP can still maintain stable adsorption performance after multiple regeneration cycles. After washing with a mixed solution of MeOH and AA, tolfenpyrad can be eluted from MMIPs via Soxhlet extraction. Thus, the synthesized MMIPs not only have strong adsorption capacity and high stability, but can also be recycled by simple methods. To investigate the potential applicability of the fabricated polymer materials, MMIPs were applied to the detection of tolfenpyrad in lettuce samples. Compared with the previously reported detection methods for tolfenpyrad, the optimal MIT-LC-MS/MS approach showed acceptable linearity and good accuracy and precision (Table 5). In addition, the synthesized imprinted polymer with excellent selectivity can be reused many times without loss of efficiency. The above results demonstrate the simplicity and convenience of the established method for the selective determination of tolfenpyrad in the actual matrices.

## 5. Conclusions

In this study, using computational simulation (Gaussian 09), 2-VP was chosen as a functional monomer to prepare MMIPs using tolfenpyrad as a template. The properties of the synthesized materials were evaluated by SEM, BET, FT-IR, XRD, TGA, and VSM characterization experiments. The efficiencies of the polymers were assessed by the adsorption isotherm test and fitted to the adsorption isothermal model. The Freundlich isothermal model indicates that the polymer has heterogeneous imprinting sites. The proposed method avoids the complex extraction steps required by the conventional solvents in dispersive and cartridge SPE modes. Moreover, the synthesized MMIPs exhibit excellent regeneration properties without any significant loss of extraction capacity. The application results of this method on the lettuce samples demonstrate that the synthesized polymers can be efficiently used for tolfenpyrad detection in real matrices such as tomatoes and cucumbers for reducing pesticide residues.

## Figures and Tables

**Figure 1 foods-12-01045-f001:**
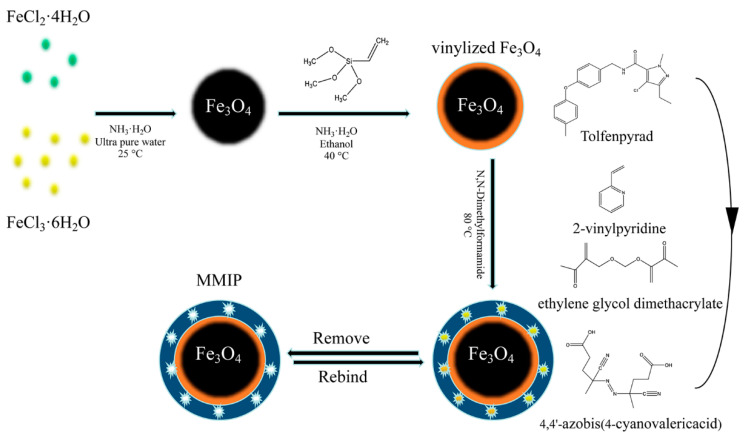
Schematic view of the preparation of MMIPs for extracting tolfenpyrad.

**Figure 2 foods-12-01045-f002:**
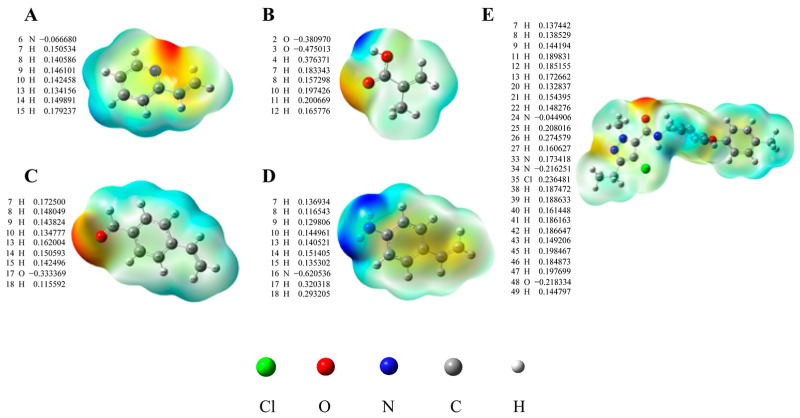
MEP distributions and surface charges of 2-VP (**A**), MAA (**B**), 4-VB (**C**), 4-VA (**D**), and tolfenpyrad (**E**) after optimization by Gaussian 09.

**Figure 3 foods-12-01045-f003:**
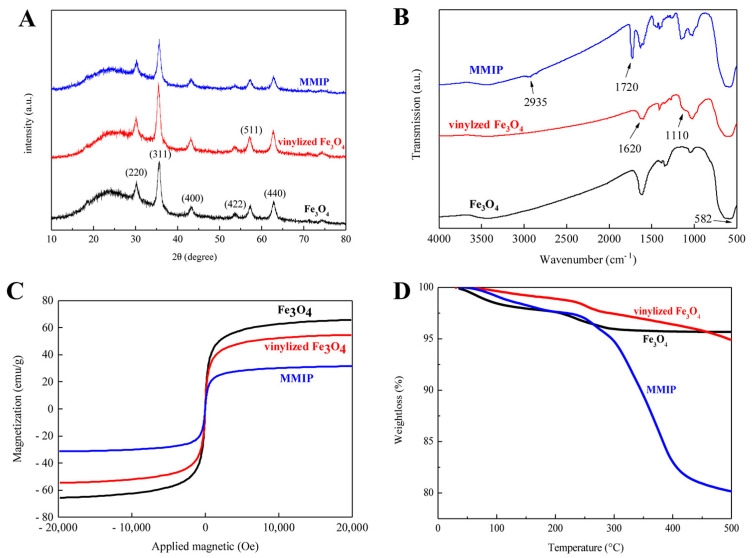
XRD patterns (**A**), FT-IR spectra (**B**), VSM (**C**), and TGA (**D**) of Fe_3_O_4_, vinylized Fe_3_O_4_, and MMIPs.

**Figure 4 foods-12-01045-f004:**
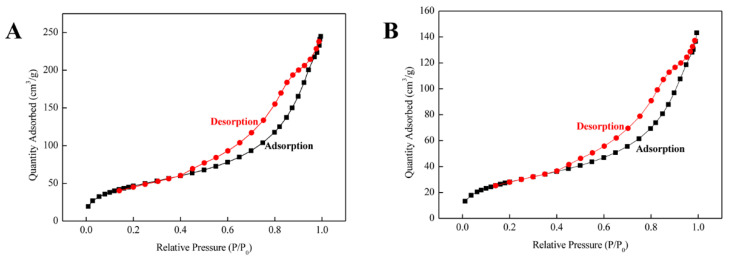
Nitrogen adsorption–desorption plots for MMIPs (**A**) and MNIPs (**B**).

**Figure 5 foods-12-01045-f005:**
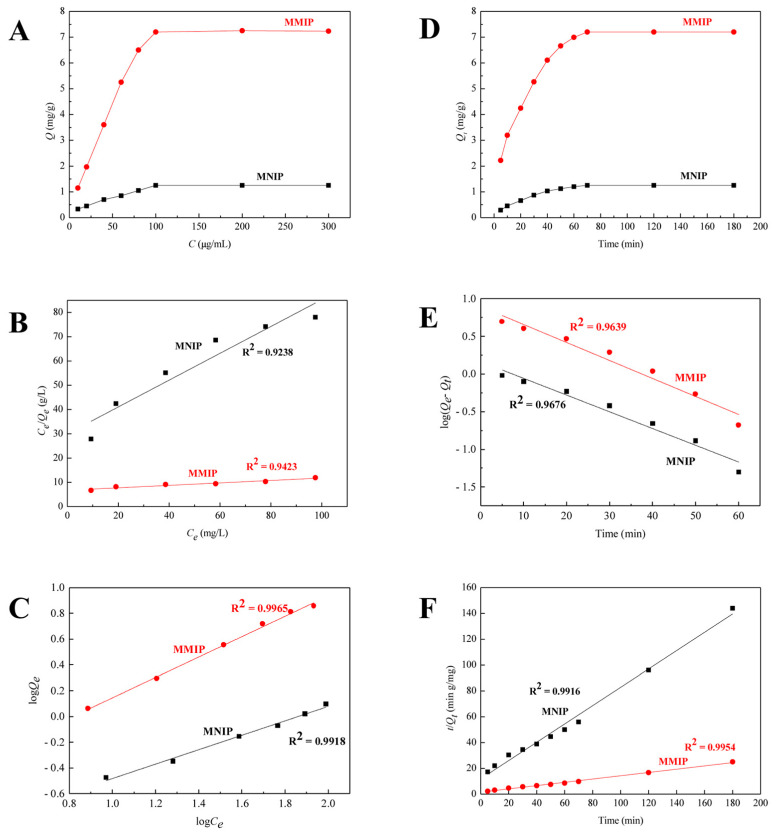
Static adsorption curves (**A**), Langmuir isothermal model (**B**), Freundlich isothermal model (**C**), dynamic adsorption curves (**D**), pseudo-first-order kinetics model (**E**), and pseudo-second-order kinetics model (**F**) of MMIPs and MNIPs.

**Figure 6 foods-12-01045-f006:**
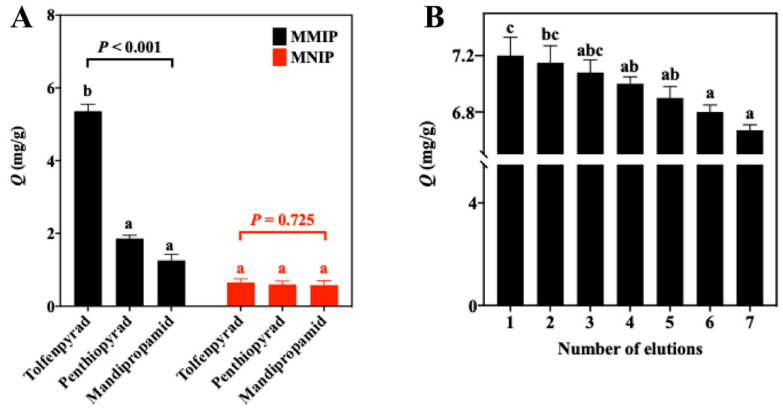
Selectivity (**A**) of MMIPs and MNIPs toward tolfenpyrad and its two structural analogs and the stability (**B**) of MMIPs (different lower-case letters indicate statistical significance between different treatments by Duncan’s multiple range test, *p* = 0.01).

**Table 1 foods-12-01045-t001:** The calculation energy of template and functional monomers.

Monomer	*E_T_* (Hartree)	*E_M_* (Hartree)	*E_P_* (Hartree)	Δ*E* (Hartree)
MAA	−1586.586463	−306.088042	−1892.677249	−0.002744
2-VP	−1586.586463	−325.208242	−1911.791631	−0.003074
4-VA	−1586.586463	−364.453837	−1951.040403	−0.000103
4-VB	−1586.586463	−422.373806	−2008.959218	−0.001051

**Table 2 foods-12-01045-t002:** Binding energies of tolfenpyrad-2-VP complexes at different imprinting ratios.

Imprinting Ratio	Binding Energy (Hartree)	Decrease in Binding Energy (Hartree)
1:1	−1912.218213	/
1:2	−2237.533439	−325.315226
1:3	−2562.844455	−325.311016
1:4	−2888.154253	−325.309798
1:5	−3213.470814	−325.316561
1:6	−3538.782370	−325.311556
1:7	−3864.099750	−325.317380
1:8	−4189.409129	−325.309379

**Table 3 foods-12-01045-t003:** Selective binding activities of MMIPs and MNIPs.

Analyte	Binding Capacity (mg/g)			
	MMIPs	MNIPs	*IF*	*SC*
Tolfenpyrad	5.36	0.65	8.25	1
Penthiopyrad	1.86	0.60	3.10	2.66
Mandipropamid	1.26	0.58	2.17	3.80

**Table 4 foods-12-01045-t004:** Recovery parameters of the present method in spiked lettuce samples.

Spiked Level (µg/kg)	Intra-Day Average Recovery, RSD (%, *n* = 5)			Inter-Day Recovery, RSD (%, *n* = 15)
	Day 1	Day 2	Day 3	
5	98.1, 4.4	98.8, 3.1	98.2, 5.2	98.3, 4.0
10	95.1, 3.7	97.8, 4.3	96.8, 4.5	96.6, 4.1
100	90.8, 1.9	90.5, 2.2	90.7, 1.4	90.6, 1.7

**Table 5 foods-12-01045-t005:** Comparison with the previously reported method for detecting tolfenpyrad in different matrices.

Sample	Method	Linear Range (mg/L)	Recovery (%)	RSD (%)	References
Tea	UPLC-MS/MS	0.001–1	81.2–106.3	≤11.8	[36]
Citrus	LC-MS/MS	0.001–0.5	80.6–113	≤9	[3]
Mandarin	LC-MS/MS	0.005–0.25	77.4–102.4	1.9–7.8	[37]
Tea	ASE-GC-MS/MS	0.005–0.04	77.9–104.8	3.6–7.5	[38]
Lettuce	MMIP-LC-MS/MS	0.005–1	90.5–98.8	1.4–5.2	This study

## Data Availability

Data are contained within the article and Appendix A.

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
