# Peer review of "Computer-Aided Prediction, Synthesis, and Characterization of Magnetic Molecularly Imprinted Polymers for the Extraction and Determination of Tolfenpyrad in Lettuce"

_foods, 2023, doi:10.3390/foods12051045_

Round 1
Reviewer 1 Report
Comments:
The manuscript “ Computer-aided prediction, synthesis and characterization of the magnetic molecularly imprinted polymers for extraction 3 and determination of tolfenpyrad in lettuce” by Kankan Zhang and colleagues synthesized magnetic molecularly imprinted polymers (MMIPs) with comprehensive characterization using SEM, TEM BET, FTIR et. The efficiencies of the polymers were determined by the adsorption isotherm test and fitted to the adsorption isothermal model. This work is highly interesting and the manuscript is also well written. Only a few small typo errors need to be addressed.
1. In line 417, typo error ‘.nd’, please correct.
2. In Figure 5, please make sure the font size in Figure 5A and B consistent.
Author Response
The manuscript “ Computer-aided prediction, synthesis and characterization of the magnetic molecularly imprinted polymers for extraction and determination of tolfenpyrad in lettuce” by Kankan Zhang and colleagues synthesized magnetic molecularly imprinted polymers (MMIPs) with comprehensive characterization using SEM, TEM BET, FTIR et. The efficiencies of the polymers were determined by the adsorption isotherm test and fitted to the adsorption isothermal model. This work is highly interesting and the manuscript is also well written. Only a few small typo errors need to be addressed.
- In line 417, typo error ‘.nd’, please correct.
Answer: Thank you for your suggestion. On line 417 of page in the previous paper, “.nd” was changed to “and”.
- In Figure 5, please make sure the font size in Figure 5A and B consistent.
Answer: Thank you for your suggestion. We have unified the font size in Figure 5A and 5B.
Reviewer 2 Report
The authors have reported a comprehensive experimental and theorytical study of binding capability of insecticide and acaricide tolfenpyrad toward a molecularly imprinted polymer. The Langmuir and Freundlich isothermal model equations have been examined via mass spectrometry.
The study has significant importance for the field of environmental chemistry and food technology, and would be of interest in the readers of Foods.
There are needed, however, few corrections of the current text.
For instance, those are:
1. The software should be deleted from the section Abstract (row 11;)
2. In Figure 4 (row 304,) there should be given coefficients of linear correlation.
Author Response
The authors have reported a comprehensive experimental and theorytical study of binding capability of insecticide and acaricide tolfenpyrad toward a molecularly imprinted polymer. The Langmuir and Freundlich isothermal model equations have been examined via mass spectrometry. The study has significant importance for the field of environmental chemistry and food technology, and would be of interest in the readers of Foods. There are needed, however, few corrections of the current text.
For instance, those are:
- The software should be deleted from the section Abstract (row 11;)
Answer: Thank you for your suggestion. We have deleted the software information in the “Abstract” section.
- In Figure 4 (row 304,) there should be given coefficients of linear correlation.
Answer: Thank you for your suggestion. We have added the coefficients of linear correlation in revised Figure 4.
Reviewer 3 Report
Foods
Review of Manuscript Number: foods-2236332
TITLE: Computer-aided prediction, synthesis and characterization of the magnetic molecularly imprinted polymers for extraction and determination of tolfenpyrad in lettuce
General Comments
This manuscript addressed the magnetic molecularly imprinted polymer (MMIP) characterization for the analytical application of tolfenpyrad in an actual agricultural sample. The proposed method and the synthesized MMIP had a good imprinted factor (IF) and selectivity of tolfenpyrad in the actual lettuce sample. The manuscript has an overall description and discussion, and I would like to suggest accepting this paper after the authors have addressed the following comments.
Major comments
(1) Re-adsorption of the target molecule into the MIP can be either at molecular recognition sites in the conformation of the MIP (specific adsorption) or nonspecific adsorption on the MIP polymer surface (nonselective) (for example, Chin. J. Chem. 2022, 40, 635 and refs cited; Chem. Sci. ., 2022, 13, 4589; Angew. Chem. Int. Ed., https://doi.org/10.1002/anie.202113528; Sep Sci Plus, 2023;6:2200081, https://doi.org/10.1002/sscp.202200081). The interaction assumed in the computer calculation (Fig. S2) applies to both the molecular recognition site and the nonselective adsorption on the surface. The good selectivity was shown in Fig. 5, it was assumed that the target molecule was incorporated into the molecular recognition site in the synthesized MMIP. I recommend inserting this point briefly in the text.
(2) Page 5 Line 224 – 234
The story of the MMIP synthesized to make the process as simple as possible was described in the Introduction section. However, the use of MMIP when applied to real samples follows the conventional LC-MS pretreatment. Is it possible to directly throw MMIP into a sample dispersion as a pretreatment, and to what extent was the influence of matrix (effect), and briefly insert it in the text?
Minor comments
(3) There seemed to be a whitewash error in the schematic in Fig. S1. Please recheck.
(4) Directly mentioning tolfenpyrad in the Abstract may confuse the reader; a note on insecticide, would be helpful.
(5) Page 4, Line 160
It is recommended that you specifically note the cleaning time for Soxhlet. Is 36 hours as described below correct?
(6) Page 8, Line 289
Please check the description Fe”3”O”4”.
I hope that my comment is useful for the improvement of the article.
Author Response
General Comments
This manuscript addressed the magnetic molecularly imprinted polymer (MMIP) characterization for the analytical application of tolfenpyrad in an actual agricultural sample. The proposed method and the synthesized MMIP had a good imprinted factor (IF) and selectivity of tolfenpyrad in the actual lettuce sample. The manuscript has an overall description and discussion, and I would like to suggest accepting this paper after the authors have addressed the following comments.
Major comments
(1) Re-adsorption of the target molecule into the MIP can be either at molecular recognition sites in the conformation of the MIP (specific adsorption) or nonspecific adsorption on the MIP polymer surface (nonselective) (for example, Chin. J. Chem. 2022, 40, 635 and refs cited; Chem. Sci. ., 2022, 13, 4589; Angew. Chem. Int. Ed., https://doi.org/10.1002/anie.202113528; Sep Sci Plus, 2023;6:2200081, https://doi.org/10.1002/sscp.202200081). The interaction assumed in the computer calculation (Fig. S2) applies to both the molecular recognition site and the nonselective adsorption on the surface. The good selectivity was shown in Fig. 5, it was assumed that the target molecule was incorporated into the molecular recognition site in the synthesized MMIP. I recommend inserting this point briefly in the text.
Answer: Thank you for your suggestion. We have added the discussion and new references in the revised text and the changes were listed as follows:
On line 390-391 of page 11in the previous paper, “The recognition sites of the MMIP can still maintain stable adsorption performance after multiple regeneration cycles.” was changed to “Re-adsorption of the target molecule into the MIP can be either at molecular recognition sites in the conformation of the MIP (specific adsorption) or nonspecific adsorption on the MIP polymer surface (nonselective) [32-35]. The interaction assumed in the computer calculation (Figure S1, Supplementary information) applies to both the molecular recognition site and the nonselective adsorption on the surface. The good selectivity (Figure 6) assumed that the target molecule was incorporated into the molecular recognition site in the synthesized MMIP. The recognition sites of the MMIP can still maintain stable adsorption performance after multiple regeneration cycles.”.
In the “References” section, “32. Xu, S.X.; He, H.; Liu, Z. New promises of advanced molecular recognition: Bioassays, single cell analysis, cancer therapy, and beyond. Chin. J. Chem. 2022, 40, 635–650.”, “33. Pang, J.L.; Li, P.F.; He, H.; Xu, S.X.; Liu, Z. Molecular imprinted polymers outperform lectin counterparts and enable more precise cancer diagnosis. Chem. Sci. 2022, 13, 4589–4597.”, “34. Li, P.F.; Pang, J.L.; Xu, S.X.; He, H.; Ma, Y.Y.; Liu, Z. A glycoform-resolved dualpmodal ratiometric immunoassay improves the diagnostic precision for hepatocellular carcinoma. Angew. Chem. Int. Ed. 2022, 61, e202113528.”, and “35. Takahashi, F.; Matsuda, K.; Nakazawa, T.; Mori, S.; Yoshida, M.; Shimizu, R.; Tatsumi, H.; Jin, J.Y. Synthesis and characterization of molecularly imprinted polymers for detection of the local anesthetic lidocaine in urine. Sep. Sci. Plus. 2023, 6, 2200081.” were added.
(2) Page 5 Line 224 – 234
The story of the MMIP synthesized to make the process as simple as possible was described in the Introduction section. However, the use of MMIP when applied to real samples follows the conventional LC-MS pretreatment. Is it possible to directly throw MMIP into a sample dispersion as a pretreatment, and to what extent was the influence of matrix (effect), and briefly insert it in the text?
Answer: Thank you for your suggestion. The MMIP in our study was used for extracting tolfenpyrad from the complex extractant of real matrix and then the concentrations of tolfenpyrad were detected by LC-MS/MS. The MMIP has specific adsorption capacity for tolfenpyrad and can effectively separate tolfenpyrad from other interferences. As the reviewer mentioned, the MMIP might be directly applied into the sample dispersion for certain compounds. However, we have not conducted this experiment in this study. In our further researches, this suggestion will be conducted in the application of MMIP for tolfenpyrad (or other pesticides). We thanks the reviewer’s kind advice.
Minor comments
(3) There seemed to be a whitewash error in the schematic in Fig. S1. Please recheck.
Answer: Thank you for your suggestion. We have deleted the whitewash error in this figure.
(4) Directly mentioning tolfenpyrad in the Abstract may confuse the reader; a note on insecticide, would be helpful.
Answer: Thank you for your suggestion. We have made some revisions as follows:
On line 10-11 of page 1 in the previous paper, “In this study, the type of functional monomer and the ratio of functional monomer to template were predicted by density function theory.” was changed to “Tolfenpyrad, a pyrazolamide insecticide, can be effectively against pests resistant to carbamate and organophosphate insecticides. In this study, a molecular imprinted polymer using tolfenpyrad as template molecule was synthesized. The type of functional monomer and the ratio of functional monomer to template were predicted by density function theory.”.
(5) Page 4, Line 160
It is recommended that you specifically note the cleaning time for Soxhlet. Is 36 hours as described below correct?
Answer: Thank you for your suggestion. The cleaning time for Soxhlet is 36 hours. On line 160 of page 4 in the previous paper, “The obtained MMIPs were washed with methanol to remove the residual reagents, then eluted for 36 h using a Soxhlet apparatus with methanol/acetic acid (9:1, v/v) until no template molecules were detected and then dried under vacuum.” was changed to “The obtained MMIPs were washed with methanol to remove the residual reagents, then eluted using a Soxhlet apparatus with methanol/acetic acid (9:1, v/v) for 36 h and then dried under vacuum.”.
(6) Page 8, Line 289
Please check the description Fe”3”O”4”.
Answer: Thank you for your suggestion. We have changed the numbers “3” and “4” as subscript in the revised paper.
Reviewer 4 Report
Chi et al. present the optimization of magnetic molecularly imprinted polymers (MMIPs) for extraction and detection of tolfenpyrad pesticide, using computer aided prediction of the functional monomer. Then they describe the characterization of the MMIPs and their use with an actual sample of spiked lettuce.
Although the research is of interest to the scientific community.
The abstract should better underline what is the aim of the research, as at present it is more clearly expressed in the title than in the abstract. At line 17, the second order kinetic model should be associated to the tolfenpyrad adsorption, which is a specific case.
Introduction
line 47. the definition of what a Molecular imprinted polymer is, and how it is formed, would improve clarity.
Line 59. The definition of what a functional monomer is and what its function is would improve clarity
Paragraph 2.2
The only instrument that has been sufficiently described is the chromatographic analysis, all the other instruments are just reported in a list. However, some more details should be added. Among the others, the wavelength and the theta range of the diffractometer should be written, the wavelength range and the type of measurement and acquisition should be added for the FT-IR. As an example of instruments description, and in general of experimental part arrangement, I suggest the paper of Zhang et al., Journal of Molecular Structure 1265 (2022) 133227
Paragraph 2.4
The picture in figure S1 could be put in the main text
Paragraph 2.5.
Both in line 163 and 164, Instead of using the term “isothermal binding experiment” which has no clear meaning, it is better to use the term ”adsorption isotherm experiment” which is more commonly used for this kind of experiments.
Please check the sign in equations (3) and (4). It should be a + instead of a -.
Paragraph 3.1
table 1 should be put in the supplementary materials, as there is no explanation on how the software works and what is the difference among PBEPBE, B3LYP and wB97XD. The only given information is the similarity between the FTIR experiment and PBEPBE, therefore the table can go to the supplementary information.
In its place, tables S1 and S2 could be put in the main text.
Paragraph 3.2
The first line should clarify what figure S3 represents (it is not written that it is a SEM image). What does the term ‘distorted’ mean? It is not clear to me.
Line 270. Which are the 4 samples? How is the core- shell structure confirmed by this experiment?
Figure 2B, a baseline correction of the FTIR data would strongly improve the readability
In general, in paragraph 3.2 only the outcome of the instruments is presented in most cases, without any discussion on the meaning of the values and their implication for the performance of adsorption of the NPs. (For example, at line 284 what does the difference in pore size between MNIP and MMIP implies?)
For this reason, I would merge paragraph 3 and 4 together in a Results and Discussion paragraph, so for every measurement an explanation is given. As there are many experiments, this would make readability simpler.
Figure 4 A and D are the same
Paragraph 3.4 .
table S5 should be put in the main text.
Paragraph 3.5
It is not clear what the paragraph is presenting, and to what the sentence at line 328 (first sentence) is referring to.
Paragraph 4
line 349-352. Smaller cavities are also evident in the MNIP. Does this mean that the size of the cavities increases the adsorption capacity?
Moreover, pictures should be taken in many areas of the sample, to statistically demonstrate that the two samples present different cavities and size of cavities
Sentence at line 396-397 is not clear
Paragraph 5 conclusions
At line 412 there is the statement that the method avoids time consuming centrifugation and filtration steps, but the sample with real lettuce was prepared using centrifugation and filtration. Please, could authors comment on this?
Author Response
Chi et al. present the optimization of magnetic molecularly imprinted polymers (MMIPs) for extraction and detection of tolfenpyrad pesticide, using computer aided prediction of the functional monomer. Then they describe the characterization of the MMIPs and their use with an actual sample of spiked lettuce.
Although the research is of interest to the scientific community, I think that in the actual form the paper does not reach the quality standards to be published in food. Therefore, I would recommend the rejection, and an eventual further resubmission after considering the following issues.
The abstract should better underline what is the aim of the research, as at present it is more clearly expressed in the title than in the abstract. At line 17, the second order kinetic model should be associated to the tolfenpyrad adsorption, which is a specific case.
Answer: Thank you for your suggestion. We have revised the “Abstract” section and the changes were listed as follows:
On line 10-24 of page 1 in the previous paper, “In this study, the type of functional monomer and the ratio of functional monomer to template were predicted by density function theory (Gaussian 09W software). Magnetic molecularly imprinted polymers (MMIPs) were synthesized using 2-vinylpyridine as functional monomer in the presence of ethylene magnetite nanoparticles at a monomer/tolfenpyrad ratio of 1:7. Scanning electron microscopy, transmission electron microscopy, nitrogen adsorption-desorption isotherms, Fourier transform infrared spectroscopy, X-ray diffractometer, thermogravimetric analyzer, and vibrational sample magnetometers are used to characterize the material. The successful synthesis of MMIPs is confirmed by the results of the characterization analysis. A pseudo-second-order kinetic model was used for the isothermal bonding experiment, and the kinetic data are in good agreement with the Freundlich isothermal model. The adsorption capacity of the polymer to the target analyte was 7.20 mg/g, indicating an excellent selective extraction capability. In addition, the adsorption capacity of the MMIPs is not significantly lost after several reuses. The MMIPs showed great analytical performance in tolfenpyrad-spiked lettuce samples, with acceptable accuracy (intra- and inter-day recoveries of 90.5%-98.8%) and precision (intra- and inter-day relative standard deviations of 1.4%-5.2%).” was changed to “Tolfenpyrad, a pyrazolamide insecticide, can be effectively against pests resistant to carbamate and organophosphate insecticides. In this study, a molecular imprinted polymer using tolfenpyrad as template molecule was synthesized. The type of functional monomer and the ratio of functional monomer to template were predicted by density function theory. Magnetic molecularly imprinted polymers (MMIPs) were synthesized using 2-vinylpyridine as functional monomer in the presence of ethylene magnetite nanoparticles at a monomer/tolfenpyrad ratio of 1:7. The successful synthesis of MMIPs is confirmed by the results of the characterization analysis by scanning electron microscopy, transmission electron microscopy, nitrogen adsorption-desorption isotherms, Fourier transform infrared spectroscopy, X-ray diffractometer, thermogravimetric analyzer, and vibrational sample magnetometers. A pseudo-second-order kinetic model fit the adsorption of tolfenpyrad, and the kinetic data are in good agreement with the Freundlich isothermal model. The adsorption capacity of the polymer to the target analyte was 7.20 mg/g, indicating an excellent selective extraction capability. In addition, the adsorption capacity of the MMIPs is not significantly lost after several reuses. The MMIPs showed great analytical performance in tolfenpyrad-spiked lettuce samples, with acceptable accuracy (intra- and inter-day recoveries of 90.5%-98.8%) and precision (intra- and inter-day relative standard deviations of 1.4%-5.2%).”.
Introduction
line 47. the definition of what a Molecular imprinted polymer is, and how it is formed, would improve clarity.
Answer: Thank you for your suggestion. We have added the description of “MIPs can be interpreted as synthetic analogues of natural biological antibody-antigen systems. It works by using a "lock and key" mechanism to selectively bind the molecules they are templated in the production process.” in the “Introduction” section.
Line 59. The definition of what a functional monomer is and what its function is would improve clarity
Answer: Thank you for your suggestion. We have added the description in the “Introduction” section. On line 59-61 of page 2 in the previous paper, “In the synthesis of MIPs, the most important task is to find appropriate functional monomers, so that they can have the maximum interaction with the template in the ground state, thus obtaining high selectivity and rebinding ability [14].” was changed to “In the synthesis of MIPs, the function of functional monomers is to form pre-polymerization complexes with templates by providing functional groups. Therefore, the most important task is to find appropriate functional monomers, so that they can have the maximum interaction with the template in the ground state, thus obtaining high selectivity and rebinding ability [14]. In the synthesis of MIPs, the function of functional monomers is to form pre-polymerization complexes with templates by providing functional groups. Therefore, the most important task is to find appropriate functional monomers, so that they can have the maximum interaction with the template in the ground state, thus obtaining high selectivity and rebinding ability [14].”.
Paragraph 2.2
The only instrument that has been sufficiently described is the chromatographic analysis, all the other instruments are just reported in a list. However, some more details should be added. Among the others, the wavelength and the theta range of the diffractometer should be written, the wavelength range and the type of measurement and acquisition should be added for the FT-IR. As an example of instruments description, and in general of experimental part arrangement, I suggest the paper of Zhang et al., Journal of Molecular Structure 1265 (2022) 133227
Answer: Thank you for your suggestion. We have modified this part and added some information of the instrument operation. On line 102-107 of page 3 in the previous paper, “A SU8100 SEM (Hitachi Ltd., Tokyo, Japan), a D8 Advance XRD (Bruker Corporation, MA, USA), a Setline STA8000 TGA (KEP Technologies, Lyon, France), a Nicolet iS5 FT-IR spectrometer (Thermo Fisher Scientific Inc., MA, USA), a Lake Shore 7404 VSM (Lake Shore Cryotronics, Inc., Ohio, USA), and a ASAP2460 BET (Micromeritics Instrument Corporation, Maryland, USA) were used for the characterization evaluations of tolfenpyrad-polymers.” was changed to “A SU8100 SEM (Hitachi Ltd., Tokyo, Japan) was used to confirm the morphology of the polymers. A D8 Advance XRD (Bruker Corporation, MA, USA), recording in the 2θ range of 20–80° at a speed of 5°/min, was applied on the determination of crystal structure. A Nicolet iS5 FT-IR spectrometer (Thermo Fisher Scientific Inc., MA, USA; spectra scanning range, 500-4000 cm−1; mode: transmission; resolution, 4 cm−1) was used to analyze the functional groups. A Setline STA8000 TGA (KEP Technologies, Lyon, France; temperature range, 25-500 °C; heating rate, 10 °C/min) was used to determine the mass loss of curves. Moreover, a Lake Shore 7404 VSM (Lake Shore Cryotronics, Inc., Ohio, USA) and a ASAP2460 BET (Micromeritics Instrument Corporation, Maryland, USA) were applied to investigate other characteristics of tolfenpyrad polymers.”.
Paragraph 2.4
The picture in figure S1 could be put in the main text
Answer: Thank you for your suggestion. We have moved Figure S1 in the main text and renamed as Figure 1.
Paragraph 2.5.
Both in line 163 and 164, Instead of using the term “isothermal binding experiment” which has no clear meaning, it is better to use the term ”adsorption isotherm experiment” which is more commonly used for this kind of experiments.
Answer: Thank you for your suggestion. On line 163 of page 4 in the previous paper, “Isothermal binding and Kinetic adsorption experiments” was changed to “Adsorption isotherm and kinetic experiments”. On line 164 of page 4 in the previous paper, “isothermal binding” was changed to “adsorption isotherm experiment”.
Please check the sign in equations (3) and (4). It should be a + instead of a -.
Answer: Thank you for your suggestion. We have revised these two equations.
Paragraph 3.1
table 1 should be put in the supplementary materials, as there is no explanation on how the software works and what is the difference among PBEPBE, B3LYP and wB97XD. The only given information is the similarity between the FTIR experiment and PBEPBE, therefore the table can go to the supplementary information. In its place, tables S1 and S2 could be put in the main text.
Answer: Thank you for your suggestion. Table 1 has been moved to the supplementary information and renamed as Table S1. In the main text, Table S1 and Table S1 has been added and renamed as Table 1 and Table 2. The details are shown in the revised paper and supplementary information.
Paragraph 3.2
The first line should clarify what figure S3 represents (it is not written that it is a SEM image). What does the term ‘distorted’ mean? It is not clear to me.
Answer: Thank you for your suggestion. We have revised this part. On line 266-267 of page 7 in the previous paper, “As shown in Figure S3 (Supplementary information), the surface of MMIPs is more distorted and rough than that of MNIPs.” was changed to “The SEM images are shown in Figure S2 (Supplementary information). The surface of MMIPs is more rough than that of MNIPs.”.
Line 270. Which are the 4 samples? How is the core- shell structure confirmed by this experiment?
Answer: Thank you for your suggestion. It should be three samples, including Fe3O4, vinylized Fe3O4, and MMIP. The core-shell structure of MMIPs can be known by the imprinted layer on the surface of the SEM image. On line 267 of page 8 in the previous paper, “all four samples” was changed to “all three samples”.
Figure 2B, a baseline correction of the FTIR data would strongly improve the readability
Answer: Thank you for your suggestion. The baseline may be improve the readability, however, the veracity of the data will be reduced. Therefore, we have not added the baseline in this paper.
In general, in paragraph 3.2 only the outcome of the instruments is presented in most cases, without any discussion on the meaning of the values and their implication for the performance of adsorption of the NPs. (For example, at line 284 what does the difference in pore size between MNIP and MMIP implies?)
For this reason, I would merge paragraph 3 and 4 together in a Results and Discussion paragraph, so for every measurement an explanation is given. As there are many experiments, this would make readability simpler.
Answer: Thank you for your suggestion. As the reviewer mentioned, a combination of Results and Discussion may be better. However, the text structure of “Foods” includes two separated “Results” and “Discussion” sections. Following this guideline, the presentation of “Results” and “Discussion” could be not combined. We thanks for the reviewer’s kind advice and we will apply this suggestion in our future papers.
Figure 4 A and D are the same
Answer: Thank you for your suggestion. We have replaced Figure 4D with the correct figure.
Paragraph 3.4 .
table S5 should be put in the main text.
Answer: Thank you for your suggestion. Table S5 has moved in the main test and renamed as Table 3.
Paragraph 3.5
It is not clear what the paragraph is presenting, and to what the sentence at line 328 (first sentence) is referring to.
Answer: Thank you for your suggestion. We have made some changes to clarify the content. Section “3.5. Applicability of the developed method” was changed to “The linearity of the established method was evaluated in the linear range of 0.005-1 mg/L. The results showed that the calibration curve of peak area-concentration was satisfactory (R2 = 0.998). Based on the respective signal to noise ratio (S/N) of 3:1 and 10:1, the limit of detection (LOD) and limit of quantification (LOQ) of the developed approach were calculated as 1.7 and 5 µg/kg, respectively. The accuracy and precision data for method validation are shown in Table 4. The intra-day recoveries of tolfenpyrad in spiked lettuce samples at three concentrations of 5, 10 and 100 µg/kg were 90.5%-98.8% with intra-day relative standard deviations (RSDs) of 1.4%-5.2%, and the inter-day recoveries were ranged from 90.6% to 98.3% with inter-day RSDs of 1.7%-4.1%.”.
Paragraph 4
line 349-352. Smaller cavities are also evident in the MNIP. Does this mean that the size of the cavities increases the adsorption capacity? Moreover, pictures should be taken in many areas of the sample, to statistically demonstrate that the two samples present different cavities and size of cavities
Answer: Thank you for your suggestion. In our study, according to the specific surface area analysis, the difference between the area size of MNIP and MMIP is not very large. More consideration is given to the adsorption difference caused by their specific cavity structure. Different methods were used to present the characteristics of MMIP and MNIP for tolfenpyrad and the results can support our conclusions. Therefore, more SEM pictures have not been taken. We thanks the reviewer’s comments again and apply it to our future research.
Sentence at line 396-397 is not clear
Answer: Thank you for your suggestion. We have deleted this sentence to avoid the vague expression.
Paragraph 5 conclusions
At line 412 there is the statement that the method avoids time consuming centrifugation and filtration steps, but the sample with real lettuce was prepared using centrifugation and filtration. Please, could authors comment on this?
Answer: Thank you for your suggestion. We tried to clarify the advantages of the MIT method and the improper description have been revised. On line of page in the previous paper, “The proposed method avoids the time-consuming centrifugation and filtration steps required by the conventional solvents in dispersive and cartridge SPE modes.” was changed to “The proposed method avoids the complex extraction steps required by the conventional solvents in dispersive and cartridge SPE modes.”.
Round 2
Reviewer 4 Report
After first revision the article has definitely improved. Only two further modifications would be necessary to improve the readability of the paper and make it suitable for publication in food:
1- Paragraph 2.2 Instrumentation. The wavelength of the XRD apparatus should be reported, as the angular range corresponds to different dimensional range depending on the system wavelength.
2- Paragraph 3.5. Authors should write that the results are correspondent to the measurement of the lettuce sample. Moreover, the fact that the detection method is LC-MS/MS should be clearly expressed. This would improve the readability of the article
Author Response
After first revision the article has definitely improved. Only two further modifications would be necessary to improve the readability of the paper and make it suitable for publication in food:
- Paragraph 2.2 Instrumentation. The wavelength of the XRD apparatus should be reported, as the angular range corresponds to different dimensional range depending on the system wavelength.
Answer: Thank you for your suggestion. We have added the wavelength of the XRD. On line 107-108 of page 3 in the previous paper, “A D8 Advance XRD (Bruker Corporation, MA, USA), recording in the 2θ range of 20–80° at a speed of 5°/min, was applied on the determination of crystal structure.” was changed to “A D8 Advance XRD (Bruker Corporation, MA, USA), recording in the 2θ range of 20–80° at a speed of 5°/min under CuKα radiation (λ = 0.15418 nm), was applied on the determination of crystal structure.”.
- Paragraph 3.5. Authors should write that the results are correspondent to the measurement of the lettuce sample. Moreover, the fact that the detection method is LC-MS/MS should be clearly expressed. This would improve the readability of the article.
Answer: Thank you for your suggestion. In the “3.5. Applicability of the developed method” section, “The developed method, a combination method of MIT pretreatment and LC-MS/MS detection, was validated and applied on determination of tolfenpyrad in lettuce samples.” was added.